# Postoperative analgesia using dezocine alleviates depressive symptoms after colorectal cancer surgery: A randomized, controlled, double-blind trial

**Peng Zhao, Zhuoxi Wu, Chunrui Li, Guiying Yang, Jinping Ding, Kai Wang, Mingming Wang, Lijuan Feng, Guangyou Duan**\*[☉]**, Hong Li**[iD]\*[☉]

Department of Anesthesiology, Second Affiliated Hospital, Army Medical University, Chongqing, China

☉ These authors contributed equally to this work.
\* dgy1986anesthesia@126.com (GYD); lh78553@163.com (HL)

## Abstract

### Background

Postoperative depression is one of the most common mental disorders in patients undergoing cancer surgery and it often delays postoperative recovery. We investigated whether dezocine, an analgesic with inhibitory effect on the serotonin and norepinephrine reuptake, could relieve postoperative depressive symptoms in patients undergoing colorectal cancer surgery.

### Methods

This randomized, controlled, single-center, double-blind trial was performed in the Second Affiliated Hospital of the Army Medical University. A total of 120 patients were randomly assigned to receive either sufentanil (1.3 μg/kg) with dezocine (1 mg/kg) (dezocine group; n = 60) or only sufentanil (2.3 μg/kg) (control group; n = 60) for patient-controlled intravenous analgesia after colorectal cancer surgery. The primary outcome was the Beck Depression Inventory score at 2 days after surgery. The secondary outcomes included the Beck Anxiety Inventory, sleep quality, and quality of recovery scores.

### Results

Compared with those in the control group, patients in the dezocine group had lower depression scores (7.3±3.4 vs. 9.9±3.5, mean difference 2.6, 95% CI: 1.4−3.9; $P$<0.001) at 2 days after surgery and better night sleep quality at the day of surgery ($P$ = 0.010) and at 1 day after the surgery ($P$<0.001). No significant difference was found in other outcomes between the two groups.

### Conclusions

Intravenous analgesia using dezocine can relieve postoperative depression symptoms and improve sleep quality in patients undergoing colorectal cancer surgery.

**Data Availability Statement:** All relevant data are within the manuscript and its Supporting Information files.

**Funding:** The study was supported by Clinical Research Projects of Second Affiliated Hospital of Army Medical University, PLA (No.2015YLC09, HL and No.2016YLC10, GD) in design of the study and collection, analysis, and interpretation of data.

**Competing interests:** The authors have declared that no competing interests exist.

## Introduction

Surgery, anesthesia, and pain are associated with the postoperative recovery of patients undergoing cancer surgery, and a growing number of studies have focused on the effect of the anesthetic or analgesic technique on these patients' postoperative outcomes[1–3]. Psychological disorders, including depression and anxiety, almost always accompany cancer diagnosis. Depression, generally considered to be one of the most important psychological disorder in patients with cancer, has a negative effect on the quality of life and decreases treatment compliance[4,5]. Studies have reported that a pre-existing depressive disorder often delays the patients' postoperative recovery[6,7]. However, few studies have investigated how perioperative intervention can alleviate postoperative depression symptoms in patients undergoing cancer surgery.

Colorectal cancer (CRC) is the fourth most commonly diagnosed cancer among both men and women in the United States[8]. In China, the incidence of CRC in 2015 ranked fifth among all cancers[9]. As previous studies reported, CRC surgery may result in persistently low physical activity levels and worse bowel symptoms, leading to a high risk of postoperative adverse psychological symptoms in these patients[10,11]. Approximately one-third of survivors experience continuing psychological problems[12]; thus, postoperative depression treatment in patients undergoing CRC surgery remains challenging.

Dezocine, a known partial μ opioid receptor agonist and κ receptor agonist, is becoming one of the most widely prescribed analgesics for postoperative pain management in China in recent years[13–15]. Several recent studies have demonstrated that dezocine can also inhibit the reuptake of norepinephrine and serotonin, both of which are major targets in depression treatment[14,16]. In particular, serotonin and norepinephrine reuptake inhibitors are first-line antidepressant drugs. Moreover, several previous studies have reported that increased serum concentrations of norepinephrine and serotonin were associated with improved depressive symptoms[17–21]. Furthermore, a recent study found that postoperative patient-controlled intravenous analgesia (PCIA) using tramadol, which has serotonin and norepinephrine reuptake inhibitory effects, can significantly decrease the postoperative depression scores[22].

Based on the above information, we hypothesized that postoperative analgesia using dezocine could provide antidepressant effect. Therefore, we investigated whether dezocine use for postoperative PCIA can relieve postoperative depression symptoms in patients undergoing CRC surgery.

## Methods

### Ethical considerations

The study protocol was approved by the Institutional Ethics Committee of the Second Affiliated Hospital of the Army Medical University (2018–029). The trial was registered prior to patient enrollment at the Chinese Clinical Trial Registry (www.chictr.org.cn; ChiCTR1800016246). Written informed consent was obtained from all patients prior to their participation. No change was made to the protocol after the trial commenced.

### Study design and patients

This was a randomized, parallel-controlled, double-blind, single-center trial performed in the Second Affiliated Hospital of the Army Medical University, Chongqing, China.

All patients were first screened in the general surgery ward and were recruited according to the inclusion and exclusion criteria 1 day before the surgery. Patients aged 18 to 70 years with an American Society of Anesthesiologists physical class II to III who were scheduled for

elective laparoscopic CRC resection under general anesthesia from May 2018 to September 2018 were recruited. Patients were excluded if they met any of the following criteria: history of known psychiatric disease; history of administration of antipsychotic drugs for any reason; taking monoamine oxidase inhibitors within 2 weeks before the surgery; inability to communicate; allergy to opioids or dezocine; history of alcohol or drug abuse; or severe cardiac, hepatic, renal, or pulmonary dysfunction. One day before surgery, eligible patients who signed a written informed consent were consecutively enrolled in the study. Patients' demographic and preoperative clinical data were recorded. Beck's Depression Inventory (BDI) and Beck's Anxiety Inventory (BAI) were applied to measure the patients' preoperative depression and anxiety scores [23,24].

## Randomization and masking

Randomization was taken the form of simple randomization procedure and performed using the sealed envelope method. A biostatistician, who was not included in data management and statistical analysis, generated random numbers (in a 1:1 ratio) using the SPSS 22.0 software (SPSS Inc., Chicago, IL). An independent assistant who was not involved in the study or data analysis prepared the allocation sequence and hid the random numbers in opaque, numbered, and sealed envelopes. During the study period, patients were recruited consecutively and were randomly assigned to receive either sufentanil with dezocine (dezocine group) or sufentanil without dezocine (control group) in the operating room according to the random number in the envelopes. A study nurse, who was independent from data collection and analysis, prepared the drugs according to the randomization sequence. Since both analgesic drugs are colorless and transparent, they cannot be distinguished based on their appearance from the PCIA mechanical pump box. After the intervention was performed according to the randomization number, the envelopes were sealed again and stored until the end of the study. The study investigators, healthcare team members, and patients were blinded to the treatment group allocation throughout the study period. Perioperative data collection and postoperative follow-up interviews during the hospitalization period were performed by study investigators.

## Anesthesia procedures

In the operating room, all patients were continuously monitored using electrocardiography, invasive blood pressure measurement, pulse oximetry, and bispectral index monitoring. Intravenous administration of metoclopramide 10 mg and dexamethasone 5 mg to prevent intraoperative stress response and postoperative nausea and vomiting before anesthesia induction. Standard sequenced anesthesia induction was performed using midazolam 0.05 mg/kg, propofol 1.5–2 mg/kg, sufentanil 0.5 to 1 ug/kg, and rocuronium bromide 0.6 mg/kg. Subsequently, endotracheal intubation was performed and controlled respiration was applied with a tidal volume of 8–10 mL/kg, respiratory frequency of 13–15/min, and inspiratory-to-expiratory ratio of 1:2 to maintain an end-tidal carbon dioxide level of 35–45 mmHg. General anesthesia maintenance was performed using propofol 0.067–0.14 mg/kg/min, remifentanil 0.2–0.8 mg/kg/min, rocuronium bromide 0.6 mg/kg/h, and inhalation anesthesia with sevoflurane. At the time of skin closure, all patients were given sufentanil 0.1 μg/kg. After sobering in the postanesthesia care unit, patients were extubated and transferred to the surgery ward. If patients reported pain $\geq$ 4 on the numerical rating scale (NRS), 5-ug boluses of sufentanil were given. After 15 min, patients' pain intensity would be assessed again. When intensity of pain by NRS was less than 4 points and the vital signs was stable, the patient would be transferred to the surgical ward.

## Analgesia interventions

Analgesia was provided to the patients at the end of surgery through PCIA using 250-mL electronic pumps. Because PCIA with sufentanil has been widely used for postoperative analgesia in China and considering that using dezocine alone might not be effective for pain control in patients undergoing CRC surgery, we used sufentanil combined with dezocine for the experimental intervention, which is a commonly used combination in clinical practice and in many previous studies[15,25–27]. According to the strategy of equivalent opioid dose, 1 ug of sufentanil equals 1 mg of dezocine[27–29]. Therefore, we set the following experimental and control interventions: patients in the dezocine group received 1.3 μg/kg sufentanil (Yichang, Hubei Fu Pharmaceutical Co. Ltd, Hubei, China) plus 1 mg/kg dezocine (Yangtze River Pharmaceutical Group Co. Ltd, Jiangsu, China) diluted in 250 mL normal saline, while those in the control group received 2.3 μg/kg sufentanil diluted in 250 mL normal saline. During the first 48 h after surgery, the PCIA pump was set at a loading dose of 2 mL with continuous background infusion speed of 4 ml/h, 1 ml each press, with a lockout interval of 15 min. If patients reported pain $\geq$ 4 on the NRS, clinicians provided additional intravenous injection of 50 mg flurbiprofen axetil. To minimize performance bias, additional analgesia within groups was administered by clinicians who were blinded to the group allocation.

## Outcome measurement

The primary outcome of the study was the BDI score at 2 days after surgery. According to the standardized process, the BDI and BAI tests were administered to patients by a study investigator at the general surgery ward. In order to minimize detection bias, the tests were administered by the same investigator who had been trained by psychology experts. The BDI is a 21-item self-report inventory that measures characteristic attitudes and symptoms of depression with items being rated on a 4-point Likert scale (range 0–3)[23,30,31]. Higher scores indicate higher depression levels.

The secondary outcomes included the BAI score at 2 days after surgery, night sleep quality at the day of surgery and 1 day after surgery, and quality of postoperative recovery at 7±2 days after surgery. The BAI consists of 21 questions with a four-point response scale (range 0–3), with higher scores indicating higher anxiety levels[24]. The Chinese versions of both the BDI and BAI have been validated in previous studies[32,33]. Sleep quality was subjectively assessed by patients as good, average, or poor. Quality of recovery was assessed using the Quality of recovery-15 items (Qor-15) test, which is a validated, self-report measure of the early postoperative health status of patients. Each item is rated on an 11-point (0–10) NRS (for positive items, 0 = "none of the time" and 10 = "all the time"; for negative items, the scoring was reversed; maximum score 150), with higher scores representing better functioning[34].

Other outcomes included the pain scores at rest and movement, which were evaluated using an 11-point NRS (where 0 indicated no pain and 10 indicated the worst possible pain) at 6, 12, 24, and 48 h after surgery. The side-effects of analgesic consumption (nausea and vomiting) and additional analgesia requirement at 48 h after surgery were also recorded. In addition, postoperative major complications and hospital stay were recorded. Hospital discharge was decided by the attending surgeons. Major complications were defined as any adverse event that occurred after the surgery and required additional intervention.

Furthermore, peripheral blood samples were collected from all patients for validation of the potential biological effects of dezocine in depression symptom improvement. Blood sampling was performed at three time points in both groups: before anesthesia induction, 24 h after surgery, and 48 h after surgery. At each time point, 2 mL of peripheral blood was collected using a

vacuum tube with heparin (for anticoagulation), and the levels of serum serotonin and norepinephrine were tested using an ELISA-kit.

## Sample size calculation

Based on our pilot investigation (n = 10), the mean BDI score 2 days after CRC surgery was 10.2 (standard deviation [SD], 4.2). Comparison of the two means of parallel variables was performed as the main analysis. The minimum clinically important difference between the experimental and control groups was set as 0.5SD, e.g., 2.1[35]. And the test statistic was based on two sample T-Test (difference). Here because the SD in experimental group is unknown, we assumed it as 3.0. Thus the sample size was calculated based on 10.2±4.2 in control group and assumed 8.1±3.0 in experimental group. With significance set at 0.05 and power set at 80%, the sample size required to detect the assumed differences was about 50 patients, calculated by using PASS 11.0 software (NCSS, LLC. Kaysville, Utah, USA). Taking into account a lost-to-follow-up rate of about 20%, we aimed to include a total of 120 patients.

## Statistical analysis

Data were analyzed according to the intention-to-treat principle and following a pre-established analysis plan. According to the original allocation, all patients who were randomized in this study for the validity of the statistical calculations. In the current study, no interim analysis was performed.

Standard descriptive statistics, such as the mean (SD) and number of patients (frequency), were used to summarize the variables. The primary outcomes, i.e., postoperative BDI scores, were compared between the groups using homogeneous variances. Mean differences with 95% CI were calculated. Pre- and intraoperative data, including age, height, weight, surgery duration, and blood loss, as well as other postoperative outcomes, including the BAI scores, PCIA consumption, early walking time, Qor-15 scores, and intensive care unit and hospital stays, were compared between the two groups using the independent-sample t test for normally distributed data. The pain scores were compared between the groups using a two independent samples nonparametric test (the Mann-Whitney test). Categorical data, including education level, sex, and sleep quality, were analyzed using the chi-square test or Fisher's exact test. Two-way repeated ANOVA with LSD multiply comparisons was applied to compare the level of serotonin and norepinephrine at different time points between the two groups. Statistical analysis was performed using SPSS version 22.0. A two-tailed $P$-value less than 0.05 was considered statistically significant.

## Results

A total of 313 patients were screened for study participation. Of these, 120 patients were enrolled in the study and randomly assigned to the dezocine (n = 60) or control group (n = 60) (Fig 1). The final visit of the last randomized patient was completed on September 24, 2018. During the study period, there were no lapses in the blinding. No participant was lost to follow-up, and all patients were included in the final intention-to-treat analyses (Fig 1). The patients' demographic, historical, and intraoperative data are summarized in Table 1. And in the study no patient was observed to report pain NRS ≥4 in the post-anesthesia care unit.

The mean postoperative BDI score in the dezocine group was significantly lower than that in the control group (7.32±3.38 vs. 9.92±3.52, $P<0.001$, Table 2). The mean difference between the two groups was 2.6 (95% CI, 1.3–3.8). No significant difference was found between the two groups in the mean postoperative BAI scores (4.6±3.0 vs. 5.5±2.7, $P = 0.110$, Table 2) or Qor-15 scores (130.9±6.5 vs. 129.2±5.4, $P = 0.138$, Table 2). In addition, as shown in Fig 2, the night

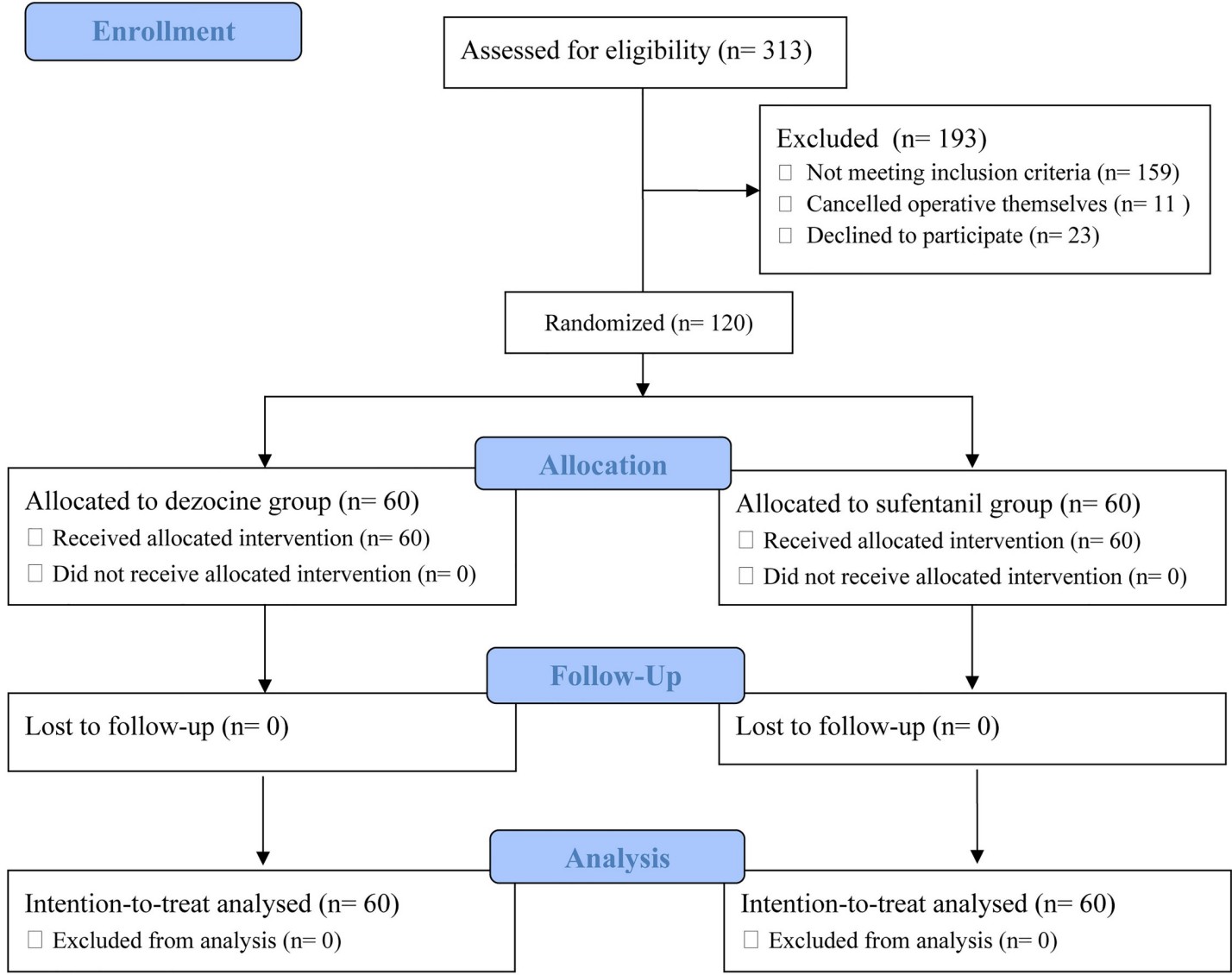

**Fig 1. CONSORT flow chart of the study.**

sleep quality was better in the dezocine group than in the control group both at the day of surgery (good 34 [56.7%], average 16 [26.7%], poor 10 [16.6%] vs. 20 [33.3%], 26 [43.3%], 14 [23.4%]; $P = 0.035$) and 1 day after the surgery (44 [73.3%], 11 [18.3%], 5 [8.4%] vs. 23 [38.3%], 28 [46.7%], 9 [15.0%]; $P<0.001$). In the dezocine group, the odds rate of good sleep at the day of surgery was 1.70 (95% CI, 1.11–2.59, $P = 0.010$) and that at 1 day after surgery was 1.91 (95% CI, 1.34–2.73, $P<0.001$).

There was no significant difference in the pain score between the two groups at 6 (rest: $P = 0.214$; movement: $P = 0.143$), 12 (rest: $P = 0.426$; movement: $P = 0.259$), 24 (rest: $P = 0.127$; movement: $P = 0.211$) or 48 h after surgery (rest: $P = 0.284$; movement: $P = 0.139$; Table 2). No significant difference was found in the total PCIA consumption or incidence of additional analgesia requirement between the two groups ($P = 0.518$ and $P = 0.094$, respectively; Table 2). In addition, there was no significant difference with respect to the incidence of nausea and vomiting, time to walking, time to resume gastrointestinal functional, length of hospital stay,

**Table 1. Patients' demographic characteristics and pre- and intraoperative data.**

| | Dezocine group(n = 60) | Control group (n = 60) |
|---|---|---|
| Male | 37(61.7%) | 31(51.7%) |
| Education level | | |
| <9 years | 17(28.3%) | 20(33.3%) |
| 9 to 12 years | 31(51.7%) | 34(56.7%) |
| > 12 years | 12(20.0%) | 6(10.0%) |
| Age; year | 56.7(11.1) | 56.0(10.9) |
| Height; cm | 160.6(9.1) | 159.9(8.3) |
| Weight; kg | 60.9(9.6) | 59.8(9.7) |
| BMI Smoking | 23.6(3.3) | 23.3(2.9) |
| No | 38(63.3%) | 41(68.3%) |
| Quit | 7(11.7%) | 5(8.3%) |
| Yes | 15(25.0%) | 14(23.3%) |
| Drinking | | |
| No | 43(71.7%) | 50(83.3%) |
| Quit | 3(5.0%) | 4(6.7%) |
| Yes | 14(23.3%) | 6(10.0%) |
| Time to confirmed diagnosis; month | 3.9(3.8) | 4.6(4.7) |
| BDI score | 5.6(3.4) | 5.3(3.1) |
| BAI score | 2.7(2.1) | 3.2(2.6) |
| ASA | | |
| II | 38(63.3%) | 34(56.7%) |
| III | 22(36.7%) | 26(43.3%) |
| Diagnosis | | |
| Colon cancer | 23(38.3%) | 23(38.3%) |
| Rectal cancer | 37(61.7%) | 37(61.7%) |
| Operative time; min | 222 (59) | 218 (51) |
| Fluid infusion volume; mL | 1791 (605) | 1958 (546) |
| Bleeding volume; mL | 199 (147) | 223 (169) |
| Urine volume; mL | 422 (364) | 416 (242) |

Values are presented as mean (SD) or number (proportion). BDI, Beck Depression Inventory; BAI, Beck Anxiety Inventory; ASA, American Society of Anesthesiologists.

or incidence of postoperative major complications during hospitalization between the two groups. ($P$ = 0.648, 0.922, 0.954, 0.471, 0.323, respectively; Table 2).

Two-way repeated ANOVA revealed a significant group effect both for the serum serotonin concentration ($P$ = 0.039) and the serum norepinephrine concentration ($P$ = 0.048) at different time points. As showed in Fig 3, there was no significant difference between the dezocine and control groups in the preoperative serum serotonin (535±138 vs. 535±149 ng/L, $P$ = 0.997) or norepinephrine levels (189±48 vs. 192±48 ng/L, $P$ = 0.751). Both the serum serotonin (535 ±142 vs. 470±139 ng/L, $P$ = 0.013) and norepinephrine levels (199±40 vs. 174±49 ng/L, $P$ = 0.002) at 1 day after surgery were higher in the dezocine than in the control group. Similarly, at 2 days after surgery, both the serum serotonin (532±147 vs. 473±127 ng/L, $P$ = 0.022) and norepinephrine levels (205±46 vs. 183±41 ng/L, $P$ = 0.008) were higher in the dezocine than in the control group.

**Table 2. Postoperative outcomes.**

| | Dezocine group(n = 60) | Control group (n = 60) | *P* value |
|---|---|---|---|
| NRS for pain at rest; score | | | |
| 6 h after surgery | 1.0(1.0) | 1.3(1.3) | 0.214 |
| 12 h after surgery | 1.0(0.9) | 1.2(1.1) | 0.426 |
| 24 h after surgery | 0.9(0.9) | 1.2(1.1) | 0.127 |
| 48 h after surgery | 0.8(0.8) | 1.0 (0.8) | 0.284 |
| NRS for pain at movement; score | | | |
| 6 h after surgery | 1.9(1.4) | 2.2(1.5) | 0.143 |
| 12 h after surgery | 2.1(1.1) | 2.2(1.3) | 0.259 |
| 24 h after surgery | 2.1(1.3) | 2.4(1.3) | 0.221 |
| 48 h after surgery | 1.8(1.0) | 2.1(1.1) | 0.139 |
| Ramsay sedation scores; score | | | |
| 6 h after surgery | 2.0(0.3) | 2.0(0.2) | 0.703 |
| 12 h after surgery | 2.0(0.0) | 2.0(0.2) | 0.560 |
| 24 h after surgery | 2.0(0.0) | 2.0(0.2) | 1.000 |
| 48 h after surgery | 2.0(0.2) | 2.0(0.2) | 0.084 |
| Nausea and vomiting | 2(3.3%) | 3(5.0%) | 0.648 |
| Total consumption of PCA; ml | 175(10) | 176(7) | 0.518 |
| Additional analgesia requirement | 1(1.7%) | 5(8.3%) | 0.094 |
| BDI; score | 7.3(3.4) | 9.9(3.5) | <0.001 |
| BAI; score | 4.6(3.0) | 5.5(2.7) | 0.110 |
| Time to walk; hour | 78.9(32.6) | 78.2(50.8) | 0.922 |
| Time to resume gastrointestinal recovery; hour | 85.0(44.4) | 84.4(67.2) | 0.954 |
| Qor-15; score | 130.9 (6.5) | 129.2(5.4) | 0.138 |
| Length of hospital stay; day | 13.8(3.9) | 14.4(5.9) | 0.471 |
| Complications | 16(26.7%) | 21(35%) | 0.323 |

Values are presented as mean (SD) or number (proportion). BDI, Beck Depression Inventory; BAI, Beck Anxiety Inventory; NRS, numeric rating scale; PCA, patient controlled analgesia; Qor-15, quality of recovery-15 items.

## Discussion

Through this randomized controlled trial, we found that postoperative intravenous analgesia using sufentanil combined with dezocine can significantly lower the depression scores

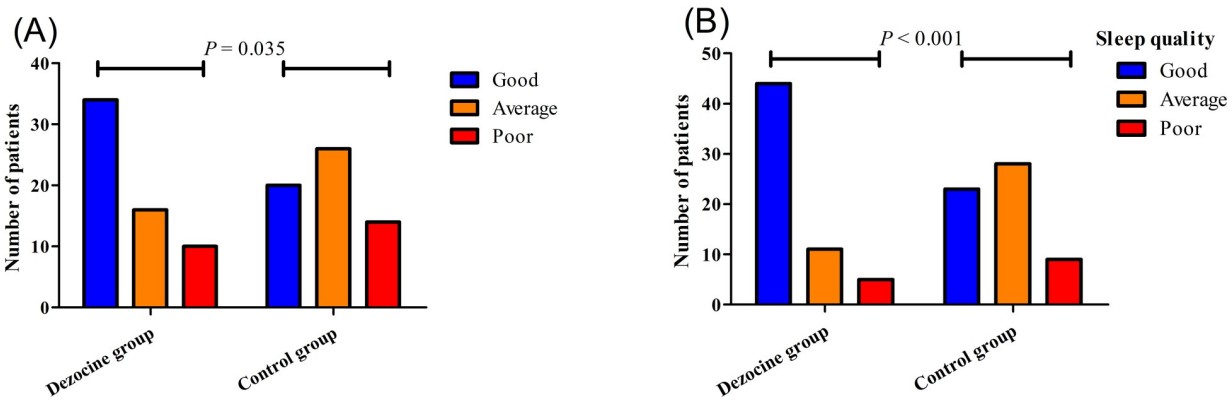

**Fig 2.** Distribution of patients' self-reported night sleep quality at the day of surgery (A) and 1 day after surgery (B).

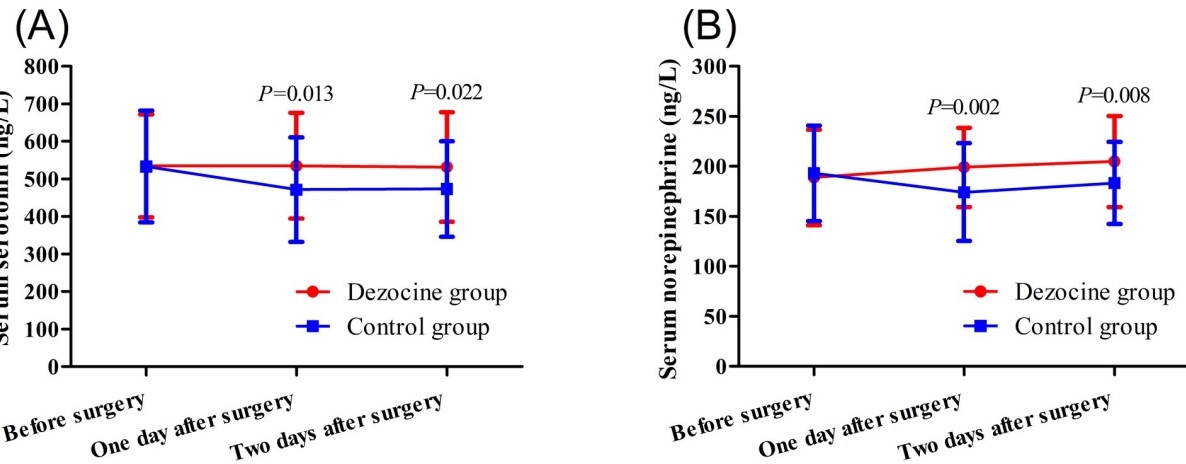

**Fig 3.** Serum serotonin (A) and norepinephrine (B) concentrations at different time points for the patients of the two groups.

compared to those in the control group at 2 days after CRC surgery. The results also showed that dezocine significantly improved the night sleep quality at the day of surgery and 1 day after surgery. Both the serum serotonin and norepinephrine levels at 1 and 2 days after surgery in the dezocine group were higher than those in the control group. No significant difference was found in the other outcomes, including postoperative anxiety, Qor-15, and pain scores between the two groups.

Dezocine is a partial µ opioid receptor agonist[27,36,37]that is theoretically approximately equipotent with morphine. Clinical studies have also proved it has the same analgesic effect as morphine[38–40]. Dezocine is becoming one of the most popular postoperative analgesics in China and is often used in combination with opioids, such as sufentanil[15,27]. In the present study, we found no significant difference in the pain scores both at rest and movement between the dezocine and control groups. The mean pain score at movement during the 48-h analgesia was less than 3 in both groups. In addition, no difference was found in the PCIA consumption or additional analgesia requirement during the 48-h follow-up. These results indicated that sufentanil combined with dezocine can provide effective postoperative analgesia in patients undergoing CRC surgery.

Our findings showed that, compared to the preoperative psychological assessment, patients experience an increase in anxiety and depressive symptoms in the early postoperative period after laparoscopic CRC surgery. This finding is consistent with that of a previous study, which found that patients experienced an increase in depressive symptoms in the early postoperative period after CRC surgery[41,42]. To the best of our knowledge, this study is the first to explore the role of PCIA using dezocine in relieving depression symptoms after laparoscopic CRC surgery. We found that patients receiving dezocine PCIA had significantly lower depression scores than those of the control group at 2 days after surgery, indicating that dezocine has the potential to relieve postoperative depression symptoms. A recent study found that postoperative patient-controlled epidural analgesia using dezocine can decrease depression scores at 3 days after cesarean delivery and increase the patients' serum serotonin concentration compared to controls without dezocine[14]. This study supports our finding that dezocine can decrease the depression scores in surgery patients, and the underlying mechanism may be its effect on the serotonin and norepinephrine reuptake. The interaction of dezocine with serotonin and norepinephrine transporters at their ligand-binding site and its concentration-

dependent inhibitory effect on the serotonin and norepinephrine reuptake have been confirmed by several studies[16,43]; however, these studies did not provide direct clinical evidence.

Previous studies have demonstrated that deficiency of monoamine neurotransmitters, such as serotonin, noradrenaline, and dopamine, in the central and peripheral nervous system is a major cause of depression[20,44]. The changes in the serum levels of monoamine neurotransmitters and their metabolites can be used as important biomarkers for diagnosis of depression [17,18]. In the current study, we explored the possible changes in the serum serotonin and norepinephrine concentrations when dezocine was intravenously administered during PCIA. Interestingly, the results showed that both the serum serotonin and norepinephrine concentrations at 1 and 2 days after surgery were higher in the dezocine group than in the control group. Another recent study also found that plasma norepinephrine and serotonin concentrations at 1 day after laparoscopic cholecystectomy were higher in participants receiving intravenous dezocine before surgery than in controls[45]. This clinical evidence further supports the interaction of dezocine with serotonin and norepinephrine transporters. Based on our results and the available evidence, we speculated that postoperative dezocine use can relieve depression symptoms in patients undergoing CRC surgery and that this effect may be associated with its ability to elevate the serum levels of serotonin and norepinephrine.

Beyond its effect in improving the psychological state, we also found that patients in the dezocine group reported better night sleep quality at the day of surgery and 1 day after surgery. In particular, the rate difference for good night sleep at 1 day after surgery reached 35%, indicating that dezocine use during PCIA may have a profound effect in improving patients' sleep quality. There are currently no available studies focusing on the role of dezocine in improving patients' sleep quality, and the underlying mechanism remains unknown. There was one previous study[46,47] showing that selective serotonin reuptake inhibitors may induce sleep quality changes. Based on the current finding, we speculated that the effect of dezocine on the serotonin and norepinephrine reuptake may contribute to its role in sleep quality improvement. Nevertheless, the current finding may extend the clinical benefit of postoperative analgesia using dezocine for patients undergoing CRC surgery.

Despite the favorable effects of dezocine use on the postoperative psychological state and sleep quality during the first 2 days after surgery, the mean QoR-15 score in the current study at postoperative 7 days was higher in the dezocine than in the control group, but without a significant difference between the groups. This result could have potentially be caused by a type II error because of the relatively small sample size. Qor-15 is a comprehensive assessment scale for postoperative recovery, which depends on many factors, including age, surgical technique, postoperative complications, and surgical care. The average hospitalization time of the patients in our study reached more than 10 days, indicating that CRC surgery requires long-term recovery. Thus, although dezocine demonstrated favorable short-term (48-h) effects on the psychological state and sleep quality, these positive effects might not persist over time. However, the improved perioperative mental state and sleep quality are certainly associated with improved postoperative recovery[48]. Therefore, postoperative analgesia using dezocine might also be effective in other surgical populations. Future studies with a larger sample size are needed to verify this finding.

Although the current study presented some interesting findings regarding postoperative analgesia using dezocine, several limitations should be considered when interpreting the results. First, dezocine was investigated as a combination analgesic, but its possible interaction with sufentanil is unclear. The effects of dezocine on postoperative depression or sleep quality when used alone remain unknown. Second, to exclude the potential interference from difference in the analgesic effect, an equivalent dose of the two analgesic strategies was selected.

Thus, the possible bias induced by the different dose of sufentanil could not be avoided in the study. Third, because postoperative analgesia only lasted 2 days in this study, we performed psychological assessment shortly after surgery. Hence, it is unknown whether the mental improvement can last a longer time, and this needs further study. Fourth, one calculation error must be noted that the calculated sample size is estimated to be 20% of the missed follow-up patients, and the total sample size should be 125 but not 120, however no patient was missed during the study and through power calculation based on the primary outcomes (9.9 ±3.5 *vs*. 7.3±3.4, n = 60) the power reached 0.984. In addition, the sleep quality in this study was assessed only by patients' subjective report. Future studies should apply an objective method, such as polysomnography, to further explore the effects of dezocine on sleep improvement.

## Conclusions

In conclusion, PCIA using dezocine combined with sufentanil can significantly reduce postoperative depression symptoms and improve sleep quality in patients undergoing CRC surgery. Postoperative analgesia using dezocine as a combination analgesic might be an optional analgesic strategy and should be recommended for patients with cancer. Further study is needed to validate these findings in greater detail and determine the effect of dezocine on the postoperative recovery of other cancer surgery patients.

## Supporting information

**S1 Checklist. CONSORT 2010 checklist of information to include when reporting a randomised trial**∗**.**
(DOC)

**S1 Data.**
(XLSX)

## Acknowledgments

We would like to thank Editage (www.editage.cn) for English language editing.

## Author Contributions

**Conceptualization:** Guangyou Duan, Hong Li.

**Data curation:** Peng Zhao.

**Formal analysis:** Guangyou Duan, Hong Li.

**Investigation:** Peng Zhao, Zhuoxi Wu, Chunrui Li, Guiying Yang, Jinping Ding, Kai Wang, Mingming Wang, Lijuan Feng.

**Methodology:** Guangyou Duan, Hong Li.

**Supervision:** Hong Li.

**Validation:** Guangyou Duan, Hong Li.

**Writing – original draft:** Peng Zhao.

**Writing – review & editing:** Peng Zhao, Guangyou Duan, Hong Li.

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
