## [Decision Letter · Decision Letter 0]

3 Jan 2020

PONE-D-19-20751

Postoperative analgesia using Dezocine may alleviate depressive symptoms after colorectal cancer surgery: A randomized controlled double-blind trial

PLOS ONE

Dear Dr. Li,

Thank you for submitting your manuscript to PLOS ONE. After careful consideration, we feel that it has merit but does not fully meet PLOS ONE’s publication criteria as it currently stands. Therefore, we invite you to submit a revised version of the manuscript that addresses the points raised during the review process.

The manuscript has been evaluated by three reviewers; their comments are available below.

The reviewers have raised a number of concerns that should be addressed in a revision. The reviewers raise concerns about the sample size calculation as well as the fact that sufentanil dosing is different between groups, and it can be argued that the difference observed may be related to this difference in dose. The reviewers request clarification regarding the primary endpoint for the trial, as well as the randomization procedure and recommend further statistical analyses.

Could you please revise the manuscript to address the items raised?

Please note that the revised manuscript will need to undergo further review, we thus cannot at this point anticipate the outcome of the evaluation process.

We would appreciate receiving your revised manuscript by Feb 16 2020 11:59PM. Please include the following items when submitting your revised manuscript:

We look forward to receiving your revised manuscript.

Kind regards,

Iratxe Puebla

Senior Manging Editor, PLOS ONE

Journal Requirements:

3. Please note that according to our submission guidelines (http://journals.plos.org/plosone/s/submission-guidelines), outmoded terms and potentially stigmatizing labels should be changed to more current, acceptable terminology.

4. In the Methods, we note that “Depression and anxiety questionnaires were applied to measure their preoperative depression and anxiety scores”. Could you please clarify what questionnaires were used? Were they the Beck Depression Inventory and Beck Anxiety Inventory?

5. Please add citations when discussing each of the questionnaires and surveys used in this work.

'Clinical Research Projects of Second Affiliated Hospital of Army Medical University (No.2015YLC09 and No. 2016YLC10). The sponsors had no role in the study design, survey process, data analysis, or manuscript preparation.'

"NO - The sponsors had no role in the study design, survey process, data analysis, or manuscript preparation.Include this sentence at the end of your statement: The funders had no role in study design, data collection and analysis, decision to publish, or preparation of the manuscript."

Please provide an amended Funding Statement that declares *all* the funding or sources of support received during this specific study (whether external or internal to your organization) as detailed online in our guide for authors at http://journals.plos.org/plosone/s/submit-now Please state what role the funders took in the study.  If any authors received a salary from any of your funders, please state which authors and which funder. If the funders had no role, please state: "The funders had no role in study design, data collection and analysis, decision to publish, or preparation of the manuscript."

7. We note that you have indicated that data from this study are available upon request. PLOS only allows data to be available upon request if there are legal or ethical restrictions on sharing data publicly. For information on unacceptable data access restrictions, please see http://journals.plos.org/plosone/s/data-availability#loc-unacceptable-data-access-restrictions.

8. Please include captions for your Supporting Information files at the end of your manuscript, and update any in-text citations to match accordingly. Please see our Supporting Information guidelines for more information: http://journals.plos.org/plosone/s/supporting-information

Reviewers' comments:

Reviewer's Responses to Questions

**Comments to the Author**

1. Is the manuscript technically sound, and do the data support the conclusions?

Reviewer #1: Yes

Reviewer #2: Yes

Reviewer #3: Partly

2. Has the statistical analysis been performed appropriately and rigorously? 

Reviewer #1: Yes

Reviewer #2: Yes

Reviewer #3: Yes

3. Have the authors made all data underlying the findings in their manuscript fully available?

Reviewer #1: Yes

Reviewer #2: Yes

Reviewer #3: No

4. Is the manuscript presented in an intelligible fashion and written in standard English?

Reviewer #1: Yes

Reviewer #2: Yes

Reviewer #3: No

5. Review Comments to the Author

Reviewer #1: 1. in the methods section, as an exclusion criteria you have introduced "known psychiatric disease". In my opinion you might also add "history of administration of anti-psychotic drugs for any reason" as these classes of medications might be used for purposes other than psychiatric disease, for instance, neurologic disorders or pain syndromes.

2. Maintenance dose for Propofol has been presented as mg/kg/h. Please present it as mg/kg/min

3. You have used BIS to monitor the depth of anesthesia. You have used both Propofol and volatile agents for anesthesia maintenance. I suppose you might suggest balanced anesthesia as the reason but what is the rationale behind it while you could use solely TIVA?

Reviewer #2: The manuscript " Postoperative analgesia using Dezocine may alleviate depressive symptoms after colorectal cancer surgery: A randomized controlled double-blind trial " presents an interesting study on the availability of dezocine in alleviating depressive symptoms in patients undergoing colorectal cancer surgery. Some specific comments with line number where applicable based on the downloaded PDF:

L-133, Are you sure randomization number and group allocation information in envelopes were stored until the end of the study? If so, how can you made the randomized control?

L-153, When was the tracheal tube extubated? In the operating room or PACU? If patients had significant pain after extubation in PACU, how did your members do? Are there any other analgesic beside the PCIA at that time?

L-227, NRS was continuous assessed at multiple time points, comparison of the two group should completed by use of repeated ANOVA analysis. Besides, statistic data should analyzed by SPSS of new version.

There are a few spelling mistakes need careful checking.

Reviewer #3: The authors conducted a randomized controlled double-blind single center trial with a parallel group design to compare sufentanil (1.3μg/kg) with dezocine (1mg/kg) to sufentanil (2.3μg/kg) without dezocine in patients with colorectal cancer surgery with respect to Beck Depression Inventory at 2 days after surgery. They concluded, that for patients undergoing colorectal cancer surgery, intravenous analgesia using dezocine can relieve postoperative depression scores.

In general, the presentation appears to be sound scientific.

I start my revision with some general comments:

• The interpretation of the study is not clear to me. Both groups receive the same dose of anastaesia, one with a mix of sufentanil and dezocine, the other with the monosubstance of sufentanil alone. So one might argue, that the effect of sufentanil does not increase linear with dose, and thus it remains unclear, whether the comparators should be recommended.

• The authors are encouraged to include an individual patient data file which may be anonymized, but let the reviewer and the reader follow the arguments.

• The trial registration does not met the international standard according to ICMJE and PLoSOne

• The sample size calculation is incorrect. Taking the assumptions the study appears to be underpowered. Taking the two tailed significance level 5%, 90% power, effect size 0.5 using a homoscedastic variance t-test it results 87/group.

• Give details of measurement of the primary endpoint, to make clear, whether detection bias can be excluded.

• Please give details of additional analgesia within groups, to make clear, whether performance bias can be excluded.

Line 52: Give the complete trial design specification, see above. Missing information e.g. no. of centers, interim analysis etc.

Line 58-60: Remove to Method section.

Line 60: Add setting of the study to Method section.

L107: write: “randomized, double-blind, single center trial with a parallel group design in a second …”

L130: Give the name of the randomization procedure, applied.

L134: The blinding remains unclear. Describe whether study drugs could be distinguished or not.

L 206: See comments about the sample size calculation above. Give the name of the statistical test used for sample size calculation.

L215: Specify, how intention to treat was implemented.

L220: t-test using homogeneous variances

L224: abnormal is not clear. Further U-Test does not overcome non-normal distribution, so the argument is weak.

As several test procedures are used, give the name of the test, where it is used with the value of the test statistic in the result section.

L240: Test for baseline characteristics are not meaningful in randomized trials. So skip the information as well as the last column in table 1.

L274/6/9: avoid “p>0.05”, give exact p-values.

L293: Avoid * notation, give exact p-values.

6. PLOS authors have the option to publish the peer review history of their article (what does this mean?). If published, this will include your full peer review and any attached files.

Reviewer #1: Yes: Samad EJ Golzari

Reviewer #2: No

Reviewer #3: No

---

## [Author Response · Author response to Decision Letter 0]

8 Feb 2020

February 8, 2020

Iratxe Puebla

Senior Managing Editor

PLOS ONE 

Dear Editor: 

We wish to re-submit the manuscript titled “Postoperative analgesia using dezocine alleviates depressive symptoms after colorectal cancer surgery: A randomized, controlled, double-blind trial.” The manuscript ID is PONE-D-19-20751.

We thank you and the reviewers for your thoughtful suggestions and insights. The manuscript has benefited from these insightful suggestions. I look forward to working with you and the reviewers to move this manuscript closer to publication in PLOS ONE.

The manuscript has been rechecked and the necessary changes have been made in accordance with the reviewers’ suggestions. All changes in the revised document are marked in red font. The responses to all comments have been prepared and are given below. In addition, we used a professional language editing service to improve the English language throughout the manuscript. 

In line with the journal’s requirements and your recommendations please find below the revised acknowledgements and funding statements:

Acknowledgements: We would like to thank Editage (www.editage.cn) for English language editing.

.

Amended Funding Statement: The study was supported by Clinical Research Projects of the Second Affiliated Hospital of Army Medical University, PLA (No. 2015YLC09 to HL and No. 2016YLC10 to GD) in design of the study andcollection, analysis, and interpretation of data.

Regarding data availability, there are no ethical or legal restrictions. The minimal anonymized data of the current study were uploaded as a Supporting Information file (S1Data).

Thank you for your consideration. I look forward to hearing from you.

Sincerely,

Hong Li 

Editor:

Response: We thank the editor for the thoughtful comments and suggestions. We have made the necessary changes to ensure the manuscript meets the style requirements of PLOS ONE, including those for file naming.

The name of the colleague or the details of the professional service that edited your manuscript.

Response: Thank you for your suggestion. We had the manuscript edited and formatted by Editage, a professional science editing service. We believe that it now meets the requirements of PLOS ONE. All relevant data and files were uploaded.

3. Please note that according to our submission guidelines (http://journals.plos.org/plosone/s/submission-guidelines), outmoded terms and potentially stigmatizing labels should be changed to more current, acceptable terminology.

Response: We have sought a professional language editing service to ensure acceptable terminology throughout the manuscript.

4. In the Methods, we note that “Depression and anxiety questionnaires were applied to measure their preoperative depression and anxiety scores”. Could you please clarify what questionnaires were used? Were they the Beck Depression Inventory and Beck Anxiety Inventory?

Response: The depression and anxiety questionnaires refer to the Beck’s Depression Inventory (BDI) and Beck’s Anxiety Inventory (BAI). The manuscript has been revised accordingly (Page 6, lines 110-111)

5. Please add citations when discussing each of the questionnaires and surveys used in this work.

Response: We have included relevant citations in the revised manuscript. 

Beck’s Depression Inventory (BDI) (Page 6, line 112; Page 9, line 173)

Beck AT, Beamesderfer A. Assessment of depression: the depression inventory. Mod Probl Pharmacopsychiatry. 1974;7(0):151–169. 

Beck’s Anxiety Inventory (BAI) (Page 6, line 112; Page 9, line 178)

Beck AT, Epstein N, Brown G, Steer RA. An inventory for measuring clinical anxiety: psychometric properties. J Consult Clin Psychol. 1988;56:893–897.

Chinese Version of Beck’s Depression Inventory (Page 9, line 179)

Sun XY, Li YX, Yu CQ, Li LM. [Reliability and validity of depression scales of Chinese version: a systematic review]. Zhonghua Liu Xing Bing Xue Za Zhi. 2017;38:110-6.

Chinese Version of Beck’s Anxiety Inventory (Page 9, line 179)

Liang Y, Wang L, Zhu J. Factor structure and psychometric properties of Chinese version of Beck Anxiety Inventory in Chinese doctors. J Health Psychol. 2018;23:657-66.

Quality of recovery-15 items (Qor-15) ( Page 9, line 185)

Bu XS, Zhang J, Zuo YX. Validation of the Chinese Version of the Quality of Recovery-15 Score and Its Comparison with the Post-Operative Quality Recovery Scale. Patient. 2016;9:251-9.

'Clinical Research Projects of Second Affiliated Hospital of Army Medical University (No.2015YLC09 and No. 2016YLC10). The sponsors had no role in the study design, survey process, data analysis, or manuscript preparation.'

"NO - The sponsors had no role in the study design, survey process, data analysis, or manuscript preparation .Include this sentence at the end of your statement: The funders had no role in study design, data collection and analysis, decision to publish, or preparation of the manuscript."

Please provide an amended Funding Statement that declares *all* the funding or sources of support received during this specific study (whether external or internal to your organization) as detailed online in our guide for authors at http://journals.plos.org/plosone/s/submit-now

Please state what role the funders took in the study. If any authors received a salary from any of your funders, please state which authors and which funder. If the funders had no role, please state: "The funders had no role in study design, data collection and analysis, decision to publish, or preparation of the manuscript."

Response: We have included the revised statements in the cover letter. 

7. We note that you have indicated that data from this study are available upon request. PLOS only allows data to be available upon request if there are legal or ethical restrictions on sharing data publicly. For information on unacceptable data access restrictions, please see http://journals.plos.org/plosone/s/data-availability#loc-unacceptable-data-access-restrictions.

Response: Thank you for your advice. There are no ethical or legal restrictions. The minimal anonymized data of the current study was uploaded as a Supporting Information file (S1 Data).

8. Please include captions for your Supporting Information files at the end of your manuscript, and update any in-text citations to match accordingly. Please see our Supporting Information guidelines for more information: http://journals.plos.org/plosone/s/supporting-information

Response: Thank you for your advice. We have revised accordingly. 

Reviewer #1:

1. in the methods section, as an exclusion criteria you have introduced "known psychiatric disease". In my opinion you might also add "history of administration of anti-psychotic drugs for any reason" as these classes of medications might be used for purposes other than psychiatric disease, for instance, neurologic disorders or pain syndromes.

Response: We thank the reviewer for the thoughtful comments and suggestions. Patients with a history of administration of antipsychotic drugs would not be included in the current study. Based on your suggestion, we have added "history of administration of antipsychotic drugs for any reason" in the exclusion criteria (Page5, line 105-106).

2. Maintenance dose for Propofol has been presented as mg/kg/h. Please present it as mg/kg/min

Response: Thank you for your suggestion. We have revised accordingly (Page 7, line 141-142).

3. You have used BIS to monitor the depth of anesthesia. You have used both Propofol and volatile agents for anesthesia maintenance. I suppose you might suggest balanced anesthesia as the reason but what is the rationale behind it while you could use solely TIVA?

Response: In China, combined intravenous and inhalation anesthesia is a common anesthesia method, which is considered to have the advantages of enhancing the anesthetic effect and reducing the dosage of anesthetics. Furthermore, combined intravenous and inhalation anesthesia have been used in many previous studies (e.g. Lan Yuan , Wei Tang , Guo-qiangFu, et al. Combining interscalene brachial plexus block with intravenous-inhalation combined anesthesia for upper extremity fractures surgery: a randomized controlled trialInt J Surg. 2014;12:1484-1488 and World Neurosurg. 2018;111:e267-e276).

Reviewer #2: 

The manuscript " Postoperative analgesia using Dezocine may alleviate depressive symptoms after colorectal cancer surgery: A randomized controlled double-blind trial " presents an interesting study on the availability of dezocine in alleviating depressive symptoms in patients undergoing colorectal cancer surgery. Some specific comments with line number where applicable based on the downloaded PDF:

L-133, Are you sure randomization number and group allocation information in envelopes were stored until the end of the study? If so, how can you made the randomized control?

Response: We agree with the reviewer that the descriptions regarding the randomization were unclear. We have revised the text to ensure clarity, as follows: “A biostatistician, who was not included in data management and statistical analysis, generated random numbers (in a 1:1 ratio) using the SPSS 22.0 software (SPSS Inc., Chicago, IL). An independent assistant who was not involved in the study or data analysis prepared the allocation sequence and hid the random numbers in opaque, numbered, and sealed envelopes. During the study period, patients were recruited consecutively and were randomly assigned to receive either sufentanil with dezocine (dezocine group) or sufentanil without dezocine (control group) in the operating room according to the random number in the envelopes. A study nurse, who was independent from data collection and analysis, prepared the drugs according to the randomization sequence. After the intervention was performed according to the randomization number, the envelopes were sealed again and stored until the end of the study. The study investigators, healthcare team members, and patients were blinded to the treatment group allocation throughout the study period. Perioperative data collection and postoperative follow-up interviews during the hospitalization period were performed by study investigators. (Pages 6-7, lines 115-131)

L-153, When was the tracheal tube extubated? In the operating room or PACU? If patients had significant pain after extubation in PACU, how did your members do? Are there any other analgesic beside the PCIA at that time? 

Response: We thank the reviewer for the comments regarding postoperative analgesia, which should be described in the manuscript. We have revised accordingly, as follows: “At the time of skin closure, all patients were given sufentanil 0.1 μg/kg. After sobering in the post-anesthesia care unit, patients were extubated and transferred to the surgery ward. If patients reported pain ≥ 4 on the numerical rating scale (NRS), 5-ug boluses of sufentanil were given (Page 7, lines 143-146)

L-227, NRS was continuous assessed at multiple time points, comparison of the two group should completed by use of repeated ANOVA analysis. Besides, statistic data should analyzed by SPSS of new version.

Response: Thank you for comment. In this study, the pain NRS scores were abnormally distributed data; thus, repeated ANOVA was not applicable. In addition, comparing the pain NRS scores at different time points is acceptable, as seen in many previous studies (e.g.: Reynolds JW, et al. Analgesic Benefit of Pectoral Nerve Block II Blockade for Open Subpectoral Biceps Tenodesis: A Randomized, Prospective, Double-Blinded, Controlled Trial. Anesth. Analg. 2019 08;129(2)), as these comparisons could reflect the potential difference in the analgesic effect between different interventions. We have reanalyzed the data using a newer version of SPSS software (SPSS 22.0) and the results were consistent with the previous (Page 12, line 229)

There are a few spelling mistakes need careful checking.

Response: We have sought a professional language editing service to improve the English language usage throughout the manuscript.

Reviewer #3: 

The authors conducted a randomized controlled double-blind single center trial with a parallel group design to compare sufentanil (1.3μg/kg) with dezocine (1mg/kg) to sufentanil (2.3μg/kg) without dezocine in patients with colorectal cancer surgery with respect to Beck Depression Inventory at 2 days after surgery. They concluded, that for patients undergoing colorectal cancer surgery, intravenous analgesia using dezocine can relieve postoperative depression scores.

In general, the presentation appears to be sound scientific.

I start my revision with some general comments:

• The interpretation of the study is not clear to me. Both groups receive the same dose of anastaesia, one with a mix of sufentanil and dezocine, the other with the monosubstance of sufentanil alone. So one might argue, that the effect of sufentanil does not increase linear with dose, and thus it remains unclear, whether the comparators should be recommended.

Response: We thank the reviewer for the thoughtful comments and suggestions. We agree with the reviewer’s opinion that the effect of sufentanil may not increase linearly with the dose. However, at present, this is unknown and needs to be compared through further research. In addition, the use of combinations (i.e. two analgesic combinations) has been compared with single drugs in many studies (Wu L, et al. Low Concentration of Dezocine in Combination With Morphine Enhance the Postoperative Analgesia for Thoracotomy.J Cardiothorac Vasc Anesth. 2015 Aug;29(4):950-4.; Guangming Yan, et al. A prospective randomized trial of intravenous ketorolac vs. acetaminophen administered with opioid patient-controlled analgesia in gynecologic surgery. BMC Anesthesiol 2018 12 19;18(1); Wang C, Li L, Shen B, et al. A multicenter randomized double-blind prospective study of the postoperative patient controlled intravenous analgesia effects of dezocine in elderly patients[J]. Int J Clin Exp Med,2014,7(3):530-539.; Shangkun Li, et al. Application of Patient-Controlled Intravenous Analgesia of Dezocine Combined With Sufentanil in Burn Patients After Surgery. Zhonghua Shao Shang Za Zhi 31 (1), 48-51. Feb 2015. PMID 25876640.; and Lu Jing, et al. Analgesic effect of morphine combined sufentanil PCIA in patients undergoing cardiac valve surgery[J]. Journal of Clinical Anesthesiology 2014-4. 

In the present study, because PCIA with sufentanil has been widely used for postoperative analgesia in China and considering that using dezocine alone might not be effective for pain control in patients undergoing CRC surgery, we used sufentanil combined with dezocine for the experimental intervention, which is a commonly used combination in clinical practice and in many previous studies. Based on this comparison, this study aimed to explore whether dezocine use during postoperative PCIA may relieve postoperative depression symptoms in patients undergoing CRC surgery.

• The authors are encouraged to include an individual patient data file which may be anonymized, but let the reviewer and the reader follow the arguments.

Response: Thank you for your advice. The minimal anonymized data of the current study were uploaded as a Supporting Information file (S 1 Data).

• The trial registration does not met the international standard according to ICMJE and PLoS One

Response: The study protocol was approved by the Institutional Ethics Committee of the Second Affiliated Hospital of the Army Medical University (2018-029). The trial was registered prior to patient enrollment at the Chinese Clinical Trial Registry (www.chictr.org.cn; ChiCTR1800016246). The registration procedures of the Chinese Clinical Trial Registry are in full compliance with the requirements of the WHO International Clinical Trial Registration Platform (WHO ICTRP) and International Committee Medical Journal Editors (ICMJE).

• The sample size calculation is incorrect. Taking the assumptions the study appears to be underpowered. Taking the two tailed significance level 5%, 90% power, effect size 0.5 using a homoscedastic variance t-test it results 87/group.

Response: We apologize for the clerical error; we have revised the text. The power should be 80%. The sample size was calculated by a statistician from the Department of medical statistics in the hospital. Based on our pilot investigation (n=10), the mean BDI score 2 days after CRC surgery was 10.2 (standard deviation [SD], 4.2). Comparison of the two means of parallel variables was performed as in the main analysis. According to the set minimum of 0.5 SD for clinically important differences between the experimental and control groups [35], we assumed that dezocine could reduce the mean BDI scores of the experimental group by 2.1 points relative to those of the control group in this study. With significance set at 0.05 and power set at 80%, the sample size required to detect the assumed differences was 50 patients, calculated using the PASS 11.0 software (NCSS, LLC. Kaysville, Utah, USA). Taking into account a lost-to-follow-up rate of about 20%, we aimed to include a total of 120 patients. 

• Give details of measurement of the primary endpoint, to make clear, whether detection bias can be excluded.

Response: According to the standardized process, the BDI and BAI tests were administered to patients by a study investigator at the general surgery ward. In order to minimize detection bias, the tests were administered by the same investigator who had been trained by psychology experts (Page 9, lines 168-171). 

• Please give details of additional analgesia within groups, to make clear, whether performance bias can be excluded.

Response: If patients reported pain ≥ 4 on the NRS, clinicians provided additional intravenous injection of 50 mg flurbiprofen axetil. To minimize performance bias, additional analgesia within groups was administered by clinicians who were blinded to the group allocation (Page 8, lines 162-165). 

Line 52: Give the complete trial design specification, see above. Missing information e.g. no. of centers, interim analysis etc.

Response: This was a randomized, parallel-controlled, double-blind, single-center trial performed in the Second Affiliated Hospital of the Army Medical University, Chongqing, China. All patients were first screened in the general surgery ward and were recruited according to the inclusion and exclusion criteria 1 day before the surgery (Page 5, lines 98-101). In the current study, no interim analysis was performed. (Page 11, lines 214-215).

Line 58-60: Remove to Method section.

Response: We have revised accordingly.

Line 60: Add setting of the study to Method section.

Response: We have revised accordingly (Page 2, lines 34-36).

L107: write: “randomized, double-blind, single center trial with a parallel group design in a second …”

Response: We have revised accordingly (Page 5, lines 98).

L130: Give the name of the randomization procedure, applied.

Response: We have revised accordingly: “Randomization was performed using the sealed envelope method. A biostatistician, who was not included in data management and statistical analysis, generated random numbers (in a 1:1 ratio) using the SPSS 22.0 software (SPSS Inc., Chicago, IL). An independent assistant who was not involved in the study or data analysis prepared the allocation sequence and hid the random numbers in opaque, numbered, and sealed envelopes. During the study period, patients were recruited consecutively and were randomly assigned to receive either sufentanil with dezocine (dezocine group) or sufentanil without dezocine (control group) in the operating room according to the random number in the envelopes. A study nurse, who was independent from data collection and analysis, prepared the drugs according to the randomization sequence. After the intervention was performed according to the randomization number, the envelopes were sealed again and stored until the end of the study (Pages 6-7, lines 115-131).

L134: The blinding remains unclear. Describe whether study drugs could be distinguished or not.

Response: Since both analgesic drugs are colorless and transparent, they cannot be distinguished based on their appearance from the PCIA mechanical pump box (Pages 6, lines 124-126).

L 206: See comments about the sample size calculation above. Give the name of the statistical test used for sample size calculation.

Response: We have added the related description in the revision manuscript. Comparison of the two means of parallel variables was performed as the main analysis. (Page 10, lines 202-203)

L215: Specify, how intention to treat was implemented.

Response: The intention-to-treat principle implies that all patients who are randomized in a clinical trial should be analyzed according to their original allocation. According to original allocation, all patients who are randomized in this study for the validity of the statistical calculations (Page 11, lines 213-214). In fact, in this study, all data assessments were completed during the patient's hospital stay. There were no patients crossing over to another treatment group or patients lost to follow-up.

L220: t-test using homogeneous variances

Response: We have revised accordingly (Page 11, lines 218-219)

L224: abnormal is not clear. Further U-Test does not overcome non-normal distribution, so the argument is weak.

As several test procedures are used, give the name of the test, where it is used with the value of the test statistic in the result section.

Response: We have described the principle of analysis in the manuscript. In fact, the pain NRS scores were abnormally distributed data after normal test. Hence, the pain scores were compared between the groups using a two independent samples nonparametric test (the Mann-Whitney U test) (Page 11, line223- 225).

L240: Test for baseline characteristics are not meaningful in randomized trials. So skip the information as well as the last column in table 1.

Response: Thank you for your comment. We have revised accordingly.

L274/6/9: avoid “p>0.05”, give exact p-values.

Response: Thank you for your comment. We have added exact P-values in the manuscript.

L293: Avoid * notation, give exact p-values.

Response: We have added exact P-values in the manuscript.

---

## [Decision Letter · Decision Letter 1]

19 Mar 2020

PONE-D-19-20751R1

Postoperative analgesia using dezocine alleviates depressive symptoms after colorectal cancer surgery: A randomized, controlled, double-blind trial

PLOS ONE

Dear Dr. Li,

Thank you for submitting your manuscript to PLOS ONE. After careful consideration, we feel that it has merit but does not fully meet PLOS ONE’s publication criteria as it currently stands. Therefore, we invite you to submit a revised version of the manuscript that addresses the points raised during the review process.

Your revised manuscript has been assessed by two of the original three reviewers. They are generally positive about the revisions, though have raised some minor concerns about the sample size/power calculation and methodology that must be addressed.

We would appreciate receiving your revised manuscript by May 02 2020 11:59PM. To enhance the reproducibility of your results, we recommend that if applicable you deposit your laboratory protocols in protocols.io, where a protocol can be assigned its own identifier (DOI) such that it can be cited independently in the future. For instructions see: http://journals.plos.org/plosone/s/submission-guidelines#loc-laboratory-protocols

We look forward to receiving your revised manuscript.

Kind regards,

Emily Chenette

Staff Editor

PLOS ONE

Reviewers' comments:

Reviewer's Responses to Questions

**Comments to the Author**

1. If the authors have adequately addressed your comments raised in a previous round of review and you feel that this manuscript is now acceptable for publication, you may indicate that here to bypass the “Comments to the Author” section, enter your conflict of interest statement in the “Confidential to Editor” section, and submit your "Accept" recommendation.

Reviewer #2: All comments have been addressed

Reviewer #3: (No Response)

2. Is the manuscript technically sound, and do the data support the conclusions?

Reviewer #2: Yes

Reviewer #3: (No Response)

3. Has the statistical analysis been performed appropriately and rigorously? 

Reviewer #2: Yes

Reviewer #3: (No Response)

4. Have the authors made all data underlying the findings in their manuscript fully available?

Reviewer #2: Yes

Reviewer #3: (No Response)

5. Is the manuscript presented in an intelligible fashion and written in standard English?

Reviewer #2: Yes

Reviewer #3: (No Response)

6. Review Comments to the Author

Reviewer #2: Thank you for resubmitting the manuscript to PLOS ONE. I think you have read the reviewer's comments carefully and made modifications. As a whole, the manuscript is well-modified. However, there are two small flaws that may need attention.First, L156-146, If patients reported pain ≥ 4 on the numerical rating 146 scale (NRS), 5-ug boluses of sufentanil were given. As sufentanil were given, how much time is needed for patient observation?Or, how do you ensure that the extra sufentanil does not affect the dosage of the PCIA? Second,though there was no significant difference with respect to the incidence of nausea and vomiting, but no antemetic were used in PCIA in both the two groups. Is this inappropriate for the patient recovery?

Reviewer #3: The sample size calculation can still not be followed. There are two reasons: a) the statement “According to the set minimum of 0.5 SD” is not clear, in particular without giving the SD. Further the test statistic used as basis for the calculation is not given.

And additionally the total sample size of 120 does not account for a 20% dropout, when 100 patients are intended. This last statement should be given as a calculation error in the limitation section.

Second the name of the randomization procedure used is still not given.

7. PLOS authors have the option to publish the peer review history of their article (what does this mean?). If published, this will include your full peer review and any attached files.

Reviewer #2: No

Reviewer #3: No

---

## [Author Response · Author response to Decision Letter 1]

27 Mar 2020

Dear Editor: 

We wish to re-submit the manuscript titled “Postoperative analgesia using dezocine alleviates depressive symptoms after colorectal cancer surgery: A randomized, controlled, double-blind trial.” The manuscript ID is PONE-D-19-20751.

We thank you and the reviewers for your thoughtful suggestions and insights. The manuscript has benefited from these insightful suggestions. I look forward to working with you and the reviewers to move this manuscript closer to publication in PLOS ONE.

The manuscript has been rechecked and the necessary changes have been made in accordance with the reviewers’ suggestions. All changes in the revised document are marked in red font. The responses to all comments have been prepared and are given below.

In line with the journal’s requirements and your recommendations, we deposit our laboratory protocols in protocols.io where a protocol can be assigned its own identifier (DOI). DOI link: http://dx.doi.org/10.17504/protocols.io.bdysi7we .

Thank you for your consideration. I look forward to hearing from you.

Sincerely,

Hong Li 

Reviewer #2: 

1. Thank you for resubmitting the manuscript to PLOS ONE. I think you have read the reviewer's comments carefully and made modifications. As a whole, the manuscript is well-modified. However, there are two small flaws that may need attention. First, L156-146, If patients reported pain ≥ 4 on the numerical rating scale (NRS), 5-ug boluses of sufentanil were given. As sufentanil were given, how much time is needed for patient observation? Or, how do you ensure that the extra sufentanil does not affect the dosage of the PCIA? 

Response: 

We thank the reviewer for the comments regarding postoperative analgesia, which should be described in the manuscript. If patients reported pain ≥ 4 on the numerical rating scale (NRS), 5-ug boluses of sufentanil were given in PACU. After 15 min, patients’ pain intensity would be assessed again. When Intensity of pain by NRS was less than 4 points and the vital signs was stable, the patient would be transferred to the surgical ward. In fact, because of intraoperative opioid use, it is rare to present pain NRS ≥4 in the PACU, and in this study there was no colorectal cancer (CRC) patient who reported pain ≥4 in PACU. Therefore, in this study PCIA dosage would not be affected by such analgesia protocol. We have revised the related descriptions in the revision manuscript accordingly. (Page7, line147-150; Page12, line245-247)

2. Second, though there was no significant difference with respect to the incidence of nausea and vomiting, but no antemetic were used in PCIA in both the two groups. Is this inappropriate for the patient recovery?

Response: We thank the reviewer for the comments regarding postoperative analgesia, which should be described in the manuscript. Actually, in our hospital all patients would be routinely injected intravenously metoclopramide 10 mg and dexamethasone 5 mg before anesthesia induction, in order to prevent intraoperative stress response and postoperative nausea. We have added this related descriptions in the revised manuscript. (Page7, line136-138) 

Reviewer #3: 

1.The sample size calculation can still not be followed. There are two reasons: a) the statement “According to the set minimum of 0.5 SD” is not clear, in particular without giving the SD. Further the test statistic used as basis for the calculation is not given.

And additionally the total sample size of 120 does not account for a 20% dropout, when 100 patients are intended. This last statement should be given as a calculation error in the limitation section.

Response: Thanks for the reviewer for the thoughtful comments and suggestions. We are sorry for the unclear descriptions regarding the sample size calculation. Based on the pilot investigation (n=10), the mean BDI scores at 2 days after CRC surgery was 10.2 (standard deviation [SD] 4.2). The minimum clinically important difference between the experimental and control groups was set as 0.5SD, e.g., 2.1. And the test statistic was based on two sample T-Test (difference). Here because the SD in experimental group is unknown, we assumed it as 3.0. Thus the sample size was calculated based on 10.2±4.2 in control group and assumed 8.1±3.0 in experimental group. With significance set at 0.05 and power set at 80%, the sample size required to detect the assumed differences was about 50 patients, calculated by using PASS 11.0 software (NCSS, LLC. Kaysville, Utah, USA). (Page6, line206-213) 

Regarding the sample size calculation, we are sorry for the error, and we have added it in the limitation section. The calculated sample size is estimated to be 20% of the missed follow-up patients, and the total sample size should be 125 but not 120, however no patient was missed in the study and through power calculation based on the primary outcomes (9.9±3.5 vs. 7.3±3.4, n=60) the power reached 0.984. (Page22, line392-396)

2. Second the name of the randomization procedure used is still not given.

Response: We take the form of simple randomization procedure. (Page6, line115-116)

---

## [Decision Letter · Decision Letter 2]

6 May 2020

Postoperative analgesia using dezocine alleviates depressive symptoms after colorectal cancer surgery: A randomized, controlled, double-blind trial

PONE-D-19-20751R2

Dear Dr. Li,

We are pleased to inform you that your manuscript has been judged scientifically suitable for publication and will be formally accepted for publication once it complies with all outstanding technical requirements.

With kind regards,

Patrice Forget

Academic Editor

PLOS ONE

Additional Editor Comments (optional):

Reviewers' comments:

Reviewer's Responses to Questions

**Comments to the Author**

1. If the authors have adequately addressed your comments raised in a previous round of review and you feel that this manuscript is now acceptable for publication, you may indicate that here to bypass the “Comments to the Author” section, enter your conflict of interest statement in the “Confidential to Editor” section, and submit your "Accept" recommendation.

Reviewer #3: All comments have been addressed

2. Is the manuscript technically sound, and do the data support the conclusions?

Reviewer #3: Yes

3. Has the statistical analysis been performed appropriately and rigorously? 

Reviewer #3: Yes

4. Have the authors made all data underlying the findings in their manuscript fully available?

Reviewer #3: Yes

5. Is the manuscript presented in an intelligible fashion and written in standard English?

Reviewer #3: Yes

6. Review Comments to the Author

Reviewer #3: I am fine with the changes, the authors made. Thus the paper is acceptable for publication from my side.

7. PLOS authors have the option to publish the peer review history of their article (what does this mean?). If published, this will include your full peer review and any attached files.

Reviewer #3: No

---

## [Editor Report · Acceptance letter]

13 May 2020

PONE-D-19-20751R2 

Postoperative analgesia using dezocine alleviates depressive symptoms after colorectal cancer surgery: A randomized, controlled, double-blind trial 

Dear Dr. Li:

I am pleased to inform you that your manuscript has been deemed suitable for publication in PLOS ONE. Congratulations! Your manuscript is now with our production department. 

With kind regards,

on behalf of

Prof. Patrice Forget 

Academic Editor

PLOS ONE